# Influence of Solid Loading on the Gel-Casting of Porous NiTi Alloys

**DOI:** 10.3390/ma15238398

**Published:** 2022-11-25

**Authors:** Ze Wang, Zhiqiang He, Bohua Duan, Xinli Liu, Dezhi Wang

**Affiliations:** School of Materials Science and Engineering, Central South University, Changsha 410083, China

**Keywords:** nickel titanium alloy, gel-casting, porous materials, solid loading, microwave sintering

## Abstract

Porous NiTi alloys are widely applied in the field of medical implant materials due to their excellent properties. In this paper, porous NiTi alloys were prepared by non-aqueous gel-casting. The influence of solid loading on the process characteristics of slurries and the microstructure and mechanical properties of sintered samples were investigated. The viscosity and the stability of slurry significantly increased with the growth of solid loading, and the slurry had better process characteristics in the solid loading range of 40–52 vol.%. Meanwhile, the porosity and average pore diameter of the sintered NiTi alloys decreased with a rise in the solid loading, while the compressive strength increased. Porous NiTi alloys with porosities of 43.3–48.6%, average pore sizes of 53–145 µm, and compressive strengths of 87–167 MPa were fabricated by gel-casting. These properties meet the requirements of cortical bone. The results suggest that the pore structure and mechanical properties of porous NiTi products produced by gel-casting can be adjusted by controlling the solid loading.

## 1. Introduction

NiTi alloys have been widely applied to the medical area as a result of their good biocompatibility, shape memory effect, excellent corrosion resistance, and strong mechanical properties [1,2]. The addition of a porous structure to dense NiTi alloy can not only avoid stress shielding effects from mismatching of the elastic moduli between natural bone and NiTi alloy but is also beneficial for the transmission of body fluids and the ingrowth of bone tissue [3,4]. Therefore, porous NiTi alloys have become the most promising materials for biomedical applications.

NiTi alloys are generally fabricated by powder metallurgy methods, including spark plasma sintering (SPS), self-propagating high-temperature synthesis (SHS), hot isostatic pressing (HIP), and conventional sintering (CS) [5,6,7,8,9,10,11]. However, these conventional methods can only obtain products with simple shapes, and subsequent machining is required to obtain products with complex shapes, which will undoubtedly result in the deterioration of the pore structure and an increase in the cost, owing to the brittleness of porous materials. Consequently, near-net-shape forming methods, such as metal injection molding (MIM), the additive manufacturing technique (AM), and Gel-casting, have been developed to prepare products with complex shapes [12,13,14,15,16]. However, MIM is not suitable for fabricating large-sized parts, and the AM is limited by its high cost and expensive equipment. Gel-casting is a novel, near-net-shape powder processing technique, which possesses the advantages of a complex shaping capability, low organic content, more uniform microstructure, low cost, and easy operation [17]. Thus, it has been used for forming ceramics and metal materials like Al_2_O_3_, AlN, SiC, ZrO_2_, FeAl, Cu alloy, and stainless steel [18,19,20,21,22,23,24].

Gel-casting is considered to have special features that can assist in the preparation of open porous metallic materials due to the convenience of adjusting the pore structure. Compared with closed-cell metallic materials [25], open porous metallic materials are more conducive to the transmission of body fluid due to open and connected pore structure; in addition, open porous metallic materials possess higher strengths than metal foams [26], so they are more suitable as implants. There are many factors that affect the porosity, pore size, and mechanical properties of porous materials fabricated by gel-casting, such as the sintering temperature and time, solid loading, and the pore forming agent and its content [27,28,29]. Among these factors, the powder volume fraction is the most important. In addition to controlling the pore structure, it can almost have a significant influence on every process in gel-casting, such as casting, curing, debinding, and sintering. It is thought that slurry with a low viscosity and high solid loading is the basis for preparing high-quality products by gel-casting. Previous research has reported the effects of solid loading on the properties of slurries and sintered specimens in the ceramic field [30,31,32]. For example, Duan [32] found that the mechanical performance of SiC-10AlN ceramic first increased with an increase in the solid content and then became obviously worse while the solid content became 42 vol.%. This occurred due to the sharp increase in the slurry’s viscosity. However, little attention has been paid to the influence of the powder volume fraction on the gel-casting process of metal powder and its properties [22]. In fact, compared with ceramic powder, metal powder always has a coarser particle size and larger density, so having appropriate solid loading is crucial for the successful implementation of metal powder gel-casting and the attainment of high-quality porous products. As far as we know, there are no systematic reports on the effect of solid loading on the process and properties of gel-casting NiTi alloys.

In this work, porous NiTi alloys were prepared through gel-casting followed by microwave sintering, and the influences of solid loading on the curing time, fluidity and stability of slurries, pore structure, and mechanical performance of sintered NiTi alloys were investigated systematically. It is believed that this research can provide a fundamental understanding of the gel-casting of porous NiTi alloy.

## 2. Experimental

### 2.1. Raw Materials

Atomized Ni powders with a grain size of −325 mesh and hydrogenated dehydrogenated Ti powders (HDH Ti) with a mean diameter of 6.37 µm were used in the experiments. Their morphologies are displayed in Figure 1. The Ti powders were irregular, while the Ni powders were nearly spherical, suggesting good flowability. Table 1 shows the constituents of the gel system used in the slurries. Solvent of N-octanol, organic monomer of hydroxyethyl methacrylate (HEMA), crosslinker of 1,6-hexanediol diacrylate (HDDA), initiator of tert-butyl peroxybenzoate (TBPB), and catalyst of N,N-dimethylaniline (DMA) were supplied by Aladdin Industrial Co., Ltd., Shanghai, China, while dispersant of silok-7456F was sourced from Shenzhen Silok Technological Co., Ltd., Shenzhen, China.

### 2.2. Experimental Process

Figure 2 shows the gel-casting preparation procedure of the NiTi alloy, which is similar to that reported in the literature [22,28,29]. Firstly, Ti and Ni powders with a 1:1 mole ratio were blended at a rate of 120 r/min for 2 h under the protection of an Ar atmosphere in a planetary ball mill (YXQM-4L, Changsha MITR Instrument and Equipment, Changsha, China). The weight ratio of zirconia ball to blended powders was 1:1. Meanwhile, a homogeneous premixed solution with a concentration of 30 wt.% was prepared by dissolving HDDA and HEMA compounds at a ratio of 1:10 in n-octanol. Secondly, some dispersant and the blended powders were mixed into the premixed solution to get a range of slurries with different powder volume fractions based on the experimental design, in which the mass of the dispersant was 1.5% of the blended powders. After ball milling at 120 r/min in the Ar atmosphere for 12 h, the appropriate contents of DMA and TBPB were added to the homogeneous slurries and stirred quickly with a glass rod for 30 s. Thirdly, the slurries were casted into polytetra fluoroethylene (PTFE) mold and put in a 70 °C drying oven (DHG-9076A, Shanghai Jinghong Experimental Equipment Co., Ltd., Shanghai, China) for 30 min to cure the gel and manufacture wet green bodies. Subsequently, demolded green bodies were dried in a drying oven for several hours at a temperature of 80 °C. Finally, these dried green bodies were put in the tubular sintering furnace (OTF-1200X, Hefei Kejing Material Technology Co., Ltd., Hefei, China) and microwave sintering furnace (HY-SZ4516, Huaye Co., Ltd., Changsha, China) for further thermal degreasing at 450 °C for 90 min and sintering at 1050 °C for 30 min, respectively. The thermal degreasing and sintering processes both took place in an Ar atmosphere to protect the specimens from oxidation.

### 2.3. Characterization

A rotary viscometer (NDJ-79, Shanghai Changji Geology Instrument Co., Ltd., Shanghai, China) was used to measure the apparent viscosities of slurries at a shear rate of 344 s^−1^. The stability of the slurries was assessed by the sedimentation test, which was performed by pouring 10 mL slurry samples into graduated 10 mL cylinders and allowing them to settle undisturbed. Subsequently, the height of sediment (H) recorded at different times was compared with the initial height (H_0_). The densities and porosities of the sintered porous NiTi samples were determined by a hydrometer (DE-120M, Daho Meter, Dongguan, China) based on the Archimedes method. The microstructures of specimens were inspected with an optical microscope, and the obtained optical micrographs were applied for the statistical analysis of the pore size by Image-Pro software. The compressive strengths were examined on an electronic universal testing machine (DDL-300, Sinotest Equipment, Changchun, China) with Φ6 mm × 9 mm sintered samples. A scanning electron microscope (SEM, MIRA3, TESCAN, Brno, Czech Republic) was employed to observe the fracture morphology of the sintered samples. The phase compositions of sintered specimens were identified by X-ray diffraction (XRD, D/Max2500, Rigaku, Tokyo, Japan) using a Cu-Kα source (λ = 0.154 nm) with a step size of 0.02°.

## 3. Results and Discussion

### 3.1. Influence of Solid Loading on the Fluidity and Stability of Slurries

The primary step of gel-casting is to obtain stable and flowable slurries with high solid loading, which directly determines the quality of the sintered body. Good fluidity allows slurries to be successfully poured into various intricate molds without introducing defects or additional stresses, while good stability is needed to guarantee the uniformity of the green body and sintered body. There are many factors, like the powder particle size and morphology, the solid loading, and the dispersant type and concentration, that are related to the performance of slurries, but solid loading is the most important factor. Figure 3 and Figure 4 show the influences of solid loading on the apparent viscosity and stability of slurries, respectively.

As seen in Figure 3, the viscosity of slurries increased with increased solid loading, and the growth rate was gradually accelerated. When solid loading increased from 30 vol.% to 40 vol.%, the viscosity only increased from 35 mPa·s to 120 mPa·s, and the fluidity of the slurries did not change much. Once solid loading exceeded 40 vol.%, the viscosity of the slurries rose rapidly and reached 680 mPa·s at a solid loading rate of 50 vol.%. Solvent in the slurry was divided into bonded solvent absorbed by the powder surface and free solvent, with the latter playing a significant role in the fluidity. With the growth of solid loading, the volume of bonded solvent increased, resulting in a decrease in free solvent and an increase in slurry viscosity. In addition, high solid loading led to a reduction in the distance between the mixed powder particles and an increasing force between the particles. This caused mutual displacement between the particles to be more difficult. At the same time, the relative reduction of the adsorbed dispersant of the powder surface also gave rise to an increase in the slurry viscosity with solid loading. Having an excessive viscosity will make it difficult for slurries to fill molds [30,31]. Generally, gel-casting systems require well-dispersed slurries with viscosities of less than 1 Pa·s. Therefore, the peak Ni-Ti powder loading for gel-casting in this system was approximately 52 vol.%.

In contrast to ceramics, it was difficult for the slurries of metal powder to achieve complete stability through space location-obstruct and/or electrostatic effects from the dispersant due to their high density and coarse size, causing sedimentation to take place inevitably. As seen from Figure 4, the sedimentation behaviors of different solid loading rates were reflected in changes in the stable height percentages with the settlement time. With the extension of time, the interface heights decreased rapidly and then stabilized. Moreover, the higher the solid loading, the higher the interface height ratio, and the shorter the settling time for stabilization. Particularly, the stable height of the slurry at a solid loading rate of 50 vol.% was associated in little change in sedimentation tests. The results indicate that high solid loading is beneficial for the improvement of slurry stability. This may be explained by Stockes formula,
(1)$$v=\frac{\rho_{T}-\rho_{0}}{18\mu }gd^{2}$$
where v is the particle sedimentation velocity, *ρ_T_* is the solid density, *ρ_0_* is the density of the liquid medium, *µ* is the viscosity, g is the gravitational constant, and d is the particle size. An increase in solid loading results in an increase in viscosity, which will reduce particle sedimentation based on the Stockes formula. Furthermore, powder sedimentation worsens the homogeneity of green bodies, which will lead to the distortion of samples during densification and an inhomogeneous microstructure. It was found in the experiment that if the interface height ratio was less than 99.5%, it was difficult to obtain high-quality products. Therefore, the time corresponding to an interface height ratio of 99.5% might be regarded as the optimal slurry retention time, making it necessary to complete the slurry curing process within this time range. When the solid loading rate increased from 30 vol.% to 50 vol.%, the optimal retention time increased from about 3 min to 20 min. This implies that slurries with high solid loading have better process operability.

### 3.2. Influence of Solid Loading on the Curing Process

In essence, curing is a kind of polymerization reaction between the crosslinking agent and organic monomer in the slurry. Its main purpose is to give the green body adequate strength for facilitating handling during subsequent processes like demolding and drying. In the experiment, once appropriate amounts of catalyst and initiator had been injected, the slurry was stirred quickly and immediately placed in a water bath at 80 °C. Meanwhile, a tes1310 thermometer temperature probe was inserted into the slurry to monitor the temperature over time, and the period from the start temperature change point to the maximum temperature transition point was regarded as the curing time [33]. The curing times of slurries with different solid loading rates are shown in Figure 5.

It is evident that with a greater solid loading rate, a shorter curing time was recorded. As the solid loading rate increased from 35 vol.% to 50 vol.%, the curing time decreased from 357 s to 225 s. On one hand, high solid loading brings about an increase in the surface area of the powder, which might accelerate the initiation of the polymerization reaction and promote the crosslinking reaction; on the other hand, with an increase in solid loading, the free solvent in the solvent decreases, thus shortening the induction period and reaction period in the curing reaction [34]. In fact, there are appropriate curing times for products with different shapes and sizes. An excessive curing speed will cause the slurry to complete the curing process before fully filling the mold. An insufficient curing speed will not only extend the operation time but will also increase the inhomogeneity of the green body, resulting in the deterioration of the quality of sintered samples. The ideal curing time is generally required to be less than the optimal retention time of the slurry in order to prevent obvious sedimentation. It should be greater than the sum of filling mold time and the experimental preparation time. When the solid loading was 35 vol.%, the optimal retention time and curing time were 300 s and 356.5 s, respectively, parameters that would fail to obtain a uniform green body. It is suggested that the solid loading of the slurry should be more than 40 vol.% in order to prepare high-quality green bodies.

### 3.3. Influence of Solid Loading on the Microstructure of Sintered NiTi Alloys

Solid loading of slurry plays an essential role in the success of NiTi alloy gel-casting. In the experiment, when the solid loading was above 52 vol.%, the viscosity of slurry was too high to be poured into the mold smoothly. Nevertheless, if the solid loading of the slurry was less than 40 vol.%, deformation, cracking, and even collapse of sintered pieces would occur due to large and non-uniform shrinkage during sintering owing to the high sedimentation rate. Porous NiTi alloy prepared by gel-casting is shown in Figure 6. Figure 7 presents the microstructures of the sintered porous NiTi samples with the powder volume fraction varying from 42 to 51 vol.%, and Figure 8 exhibits the corresponding pore diameter distribution.

Figure 7 shows that the interconnected pores were evenly dispersed in the NiTi alloys, and contiguous grains interconnected to form a strong skeleton. The pore diameter was distributed in a wide range, and there were some small pores in these skeleton struts. At a solid loading rate of 42 vol.%, pore diameter varied from 0 to 360 µm and was mainly concentrated in the range of 80–160 µm (seen in Figure 8). There are three main sources of pores in porous NiTi alloys fabricated by gel-casting. One is from occupation of the original solvent and organic reagent in slurries, where the pore diameter is associated with the spaces among particles. Another is Kirkendall pores resulting from non-uniform diffusion between the Ni and Ti atoms. The size of these pores is related to the size of the Ni powders. The third type occurs due to the formation of agglomerations inherited from the suspension. Such pores cannot be ignored when the solid loading is high. The obtained wide pore diameter distribution structure has broad application prospects in medical devices. Small pores are suitable for blood and nutrition transmission, while large ones can be used for tissue ingrowths. It is also evident that the microstructures of pores in porous NiTi alloys are greatly affected by the original solid loading. Obvious shrinkage of the area of pores occurred with increased solid loading. Figure 9 shows the impact of solid loading on the porosity and average pore size of the sintered specimens.

Figure 8 and Figure 9 show that, as the solid loading rate increase, the pore diameter distribution became narrow, and both the porosity and the average pore diameter presented downward tendencies. As the solid loading rate increased from 42 vol.% to 51 vol.%, the porosity, average pore size and maximum pore size varied from 48.62%, 145 µm, and 360 µm to 43.28%, 53 µm, and 180 µm, respectively. The reason for this may be ascribed to the reduction of original solvent usage and the spaces among the particles. Furthermore, high solid loading means that the powders are more compact, which promotes the densification of powder during sintering and further reduces the porosity and pore size. The results reveal that the pore structure of porous NiTi alloys prepared by gel-casting could be tailored by controlling the solid loading of slurries.

Figure 10 shows the XRD patterns of Ni-Ti blended powders and NiTi alloy sintered from slurry with a solid loading rate of 51 vol.%.

There was a TiD_1.5_ phase in the Ni-Ti blended powders, because Ti powder used in this work was obtained by the hydrogenation dehydrogenation method. NiTi, Ni_3_Ti, and Ti_2_Ni phases were detected, as shown in Figure 10b, illustrating the reactions between elemental Ni and Ti powders that occurred during sintering densification. The results are consistent with other reports [6,10,13,29]. Meanwhile, it seems that the solid loading rate has no relationship with the phase composition of specimens, because the XRD patterns of specimens sintered from slurries with various solid loading rates were almost the same as those presented in Figure 10b.

### 3.4. Influence of Solid Loading on the Mechanical Strength of Sintered NiTi Alloys

The compressive stress–strain curves of sintered NiTi specimens with different solid loading rates are shown in Figure 11, and the corresponding compressive strengths obtained from these curves are shown in Figure 12.

As seen in Figure 11, if the initial stage, which may be associated with the procedure errors and experimental instruments, is ignored, the compression process of porous NiTi alloys can classified into three stages. The first is the linear elastic deformation stage, in which pore walls undergo elastic compression or bending deformation and the strain increases linearly with the stress. The second is the plastic yield deformation stage, in which plastic deformation exists selectively around the micropores because of stress concentration and then spreads to all of the pore walls accompanied by rapid crack propagation. The third is a densification and rupture stage, in which pore walls collapse and the fracture of specimens occurs [29,35]. Generally speaking, materials with a higher porosity have reduced mechanical properties [36]. As can be seen from Figure 12, the compressive strength of sintered NiTi specimens increased from 87 MPa to 167 MPa with the growth of solid loading from 42 vol.% to 51 vol.%. These results are similar to the properties of natural cortical bone (100–230 MPa) [37]. This mainly contributes to the low porosity of sintered specimens resulting from high solid loading in slurry. Accelerated densification due to the shorter diffusion path among atoms under the condition of high powder contact might also be responsible for the strength promotion.

Figure 13 indicates the fracture morphology of sintered porous NiTi alloys with varying solid loading rates.

It is obvious that porous NiTi alloys showed representative brittle fractures, caused by the presence of large numbers of pores. Meanwhile, a similar change trend and pore structure phenomenon with solid loading were also observed, as shown in Figure 7. Many sintering necks existed, and pore structures were relatively homogeneous and interconnected, suggesting that porous NiTi alloys produced by gel-casting have great mechanical properties.

## 4. Conclusions

Porous NiTi alloys were successfully manufactured by gel-casting with HDDA-HEMA used as the gel system. The relation between solid loading through the gel-casting process and the pore structure and mechanical properties of sintered specimens were investigated in detail. The conclusions can be described as follows:(1)Solid loading should be controlled within the range of 40–52 vol.% in order to obtain porous high-quality NiTi pieces.(2)An increase in the solid loading of slurries can accelerate the curing of the gel system.(3)With increased solid loading, the porosity and average pore size of sintered specimens decreased correspondingly. The pore structures of sintered specimens can be tailored by adjusting the solid loading.(4)Porous NiTi alloys prepared by gel-casting showed brittle fracture morphologies. Their compressive strengths increased as the solid loading increased.(5)Porous NiTi alloys with porosities of 43.3–48.6%, average pore diameters of 53–145 µm, and compressive strengths of 87–167 MPa were prepared by gel-casting. These properties meet the requirements of cortical bone.

## Figures and Tables

**Figure 1 materials-15-08398-f001:**
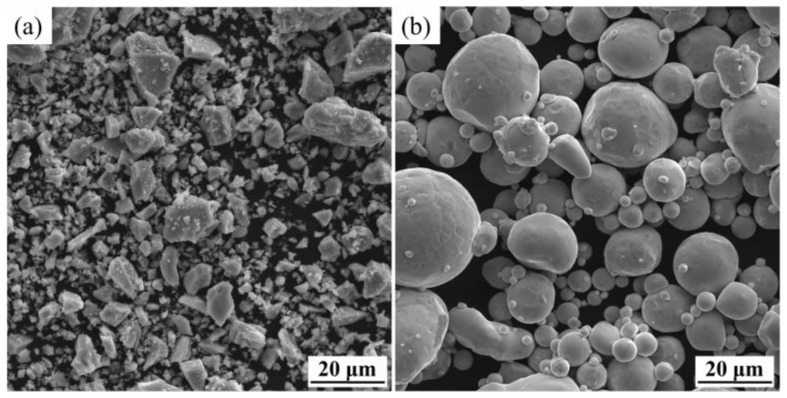
SEM images of raw powders: (**a**) HDH Ti; (**b**) Ni.

**Figure 2 materials-15-08398-f002:**
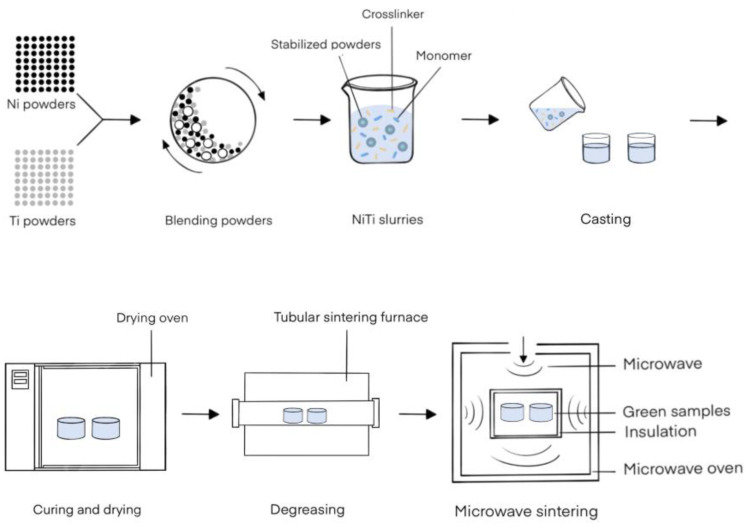
Gel-casting preparation procedure used for the NiTi alloy.

**Figure 3 materials-15-08398-f003:**
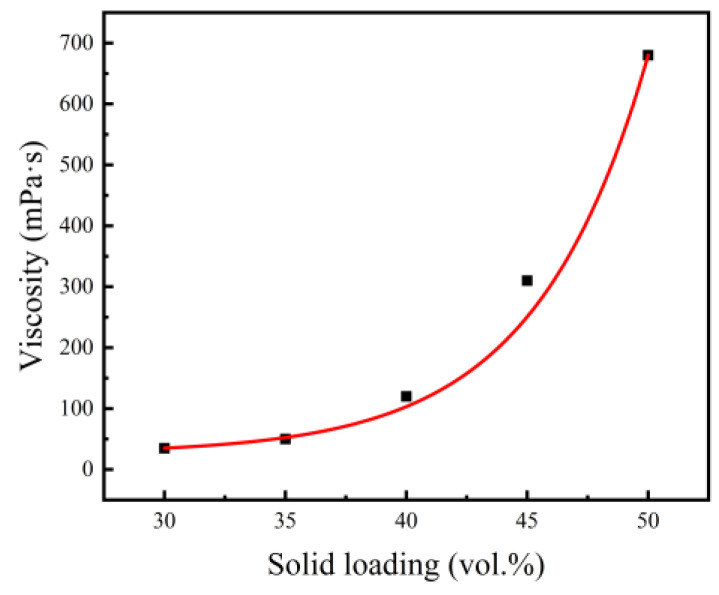
Influence of solid loading on the apparent viscosity of slurries.

**Figure 4 materials-15-08398-f004:**
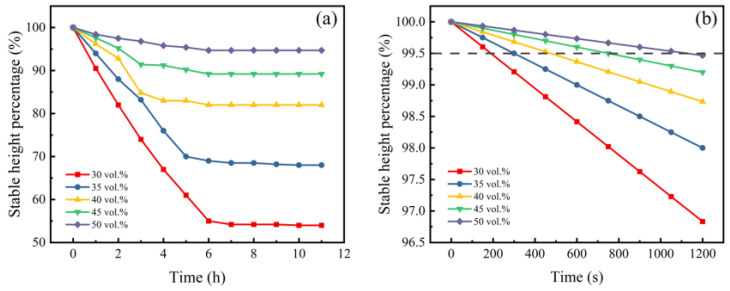
Influence of solid loading on the stability of slurries (Changes of the stable height percentages with the settlement time: (**a**) 0–11 h; (**b**) 0–1200 s).

**Figure 5 materials-15-08398-f005:**
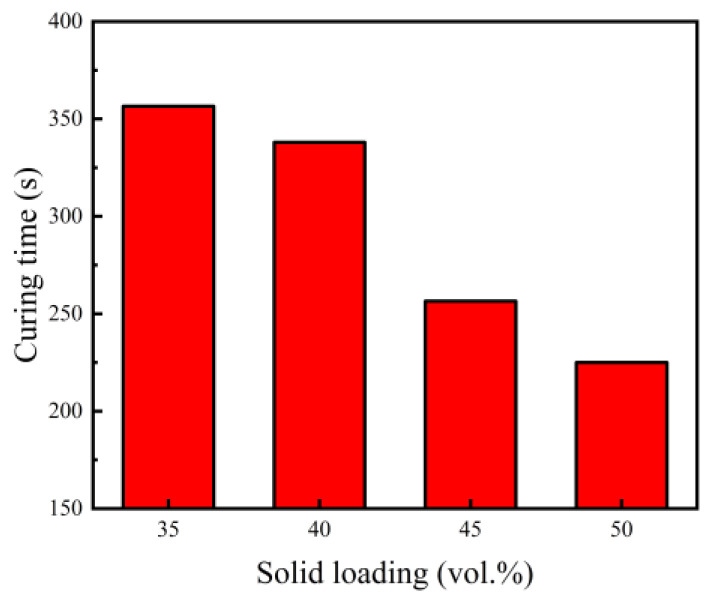
Effect of solid loading on the curing time.

**Figure 6 materials-15-08398-f006:**
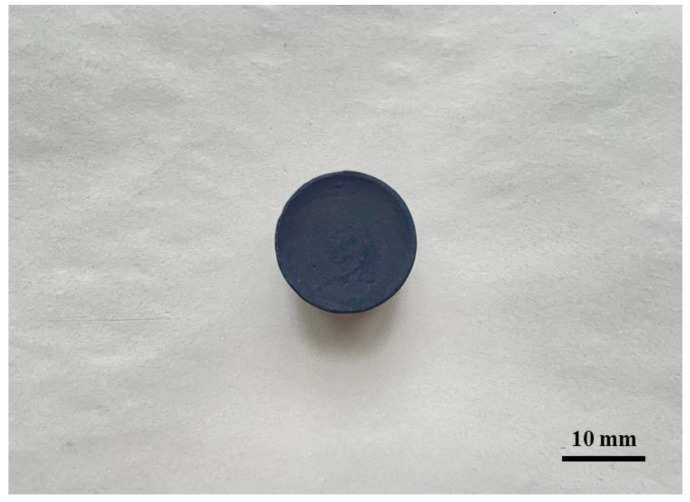
Photograph of a porous NiTi specimen sintered at 1050 °C for 30 min.

**Figure 7 materials-15-08398-f007:**
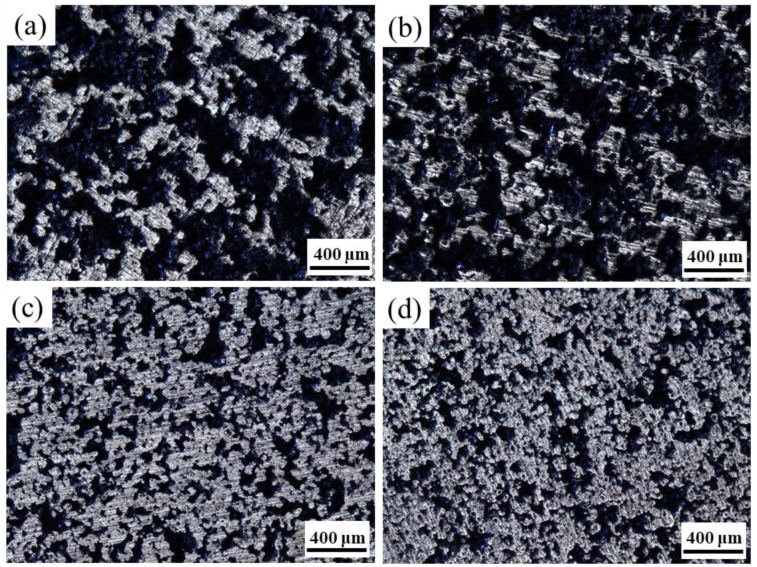
Microstructure of sintered porous NiTi alloys with different solid loading rates: (**a**) 42 vol.%; (**b**) 45 vol.%; (**c**) 48 vol.%; (**d**) 51 vol.%.

**Figure 8 materials-15-08398-f008:**
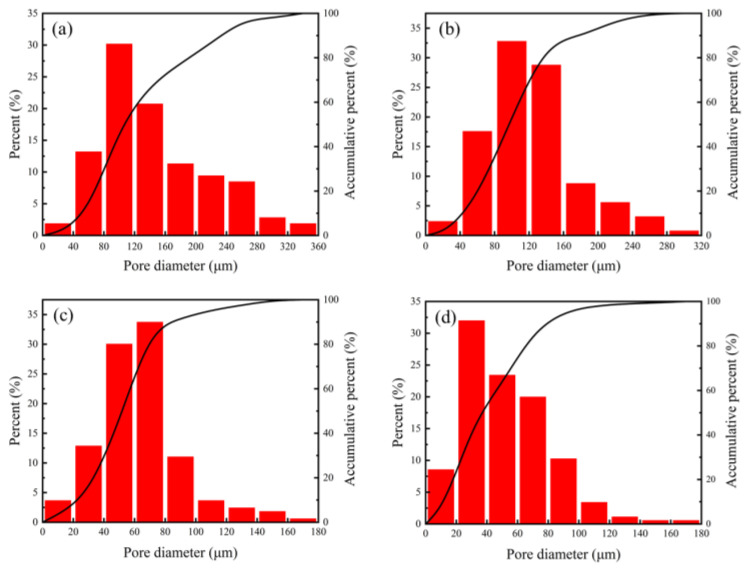
The pore diameter distribution of sintered porous NiTi alloys with different solid loading rates: (**a**) 42 vol.%; (**b**) 45 vol.%; (**c**) 48 vol.%; (**d**) 51 vol.%.

**Figure 9 materials-15-08398-f009:**
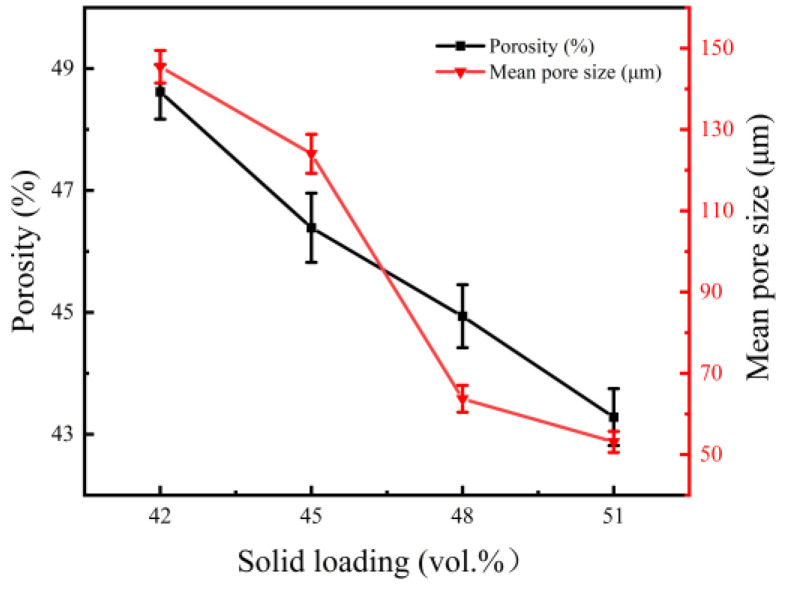
Influence of solid loading on the porosity and average pore dimeter of the sintered bodies.

**Figure 10 materials-15-08398-f010:**
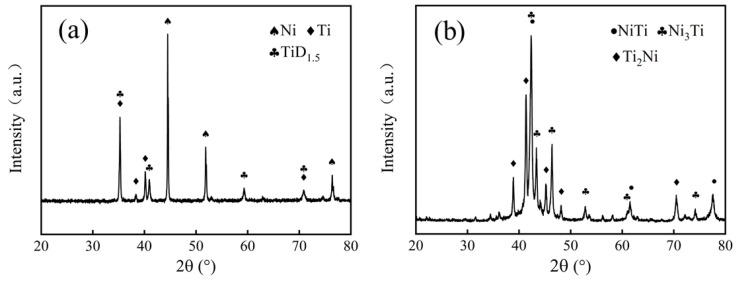
XRD patterns of (**a**) Ni-Ti blended powders and (**b**) NiTi alloy.

**Figure 11 materials-15-08398-f011:**
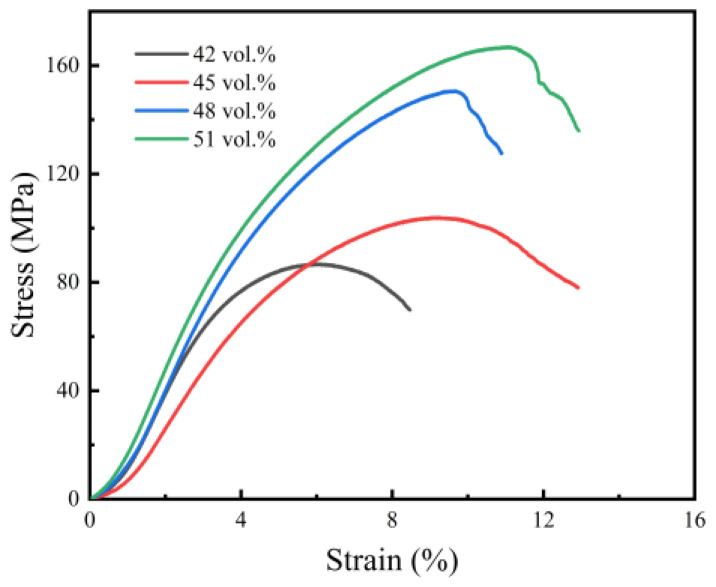
Stress–strain curves of sintered bodies.

**Figure 12 materials-15-08398-f012:**
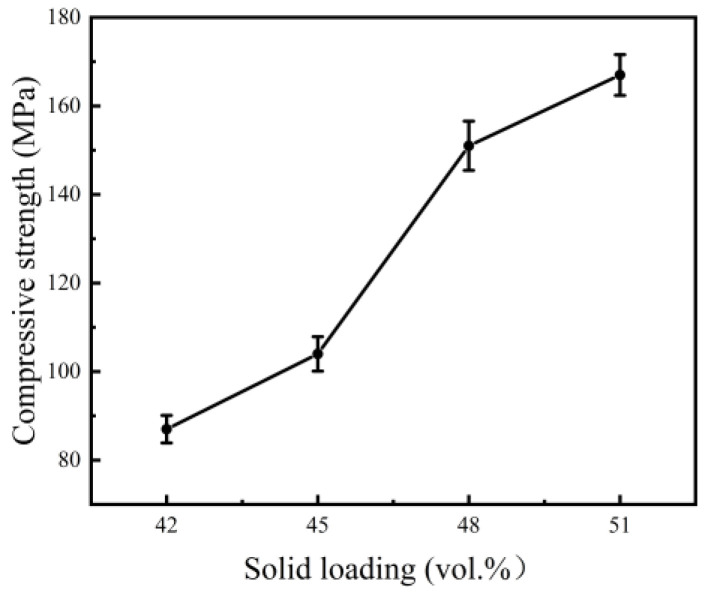
The compressive strengths of sintered specimens with various solid loading rates.

**Figure 13 materials-15-08398-f013:**
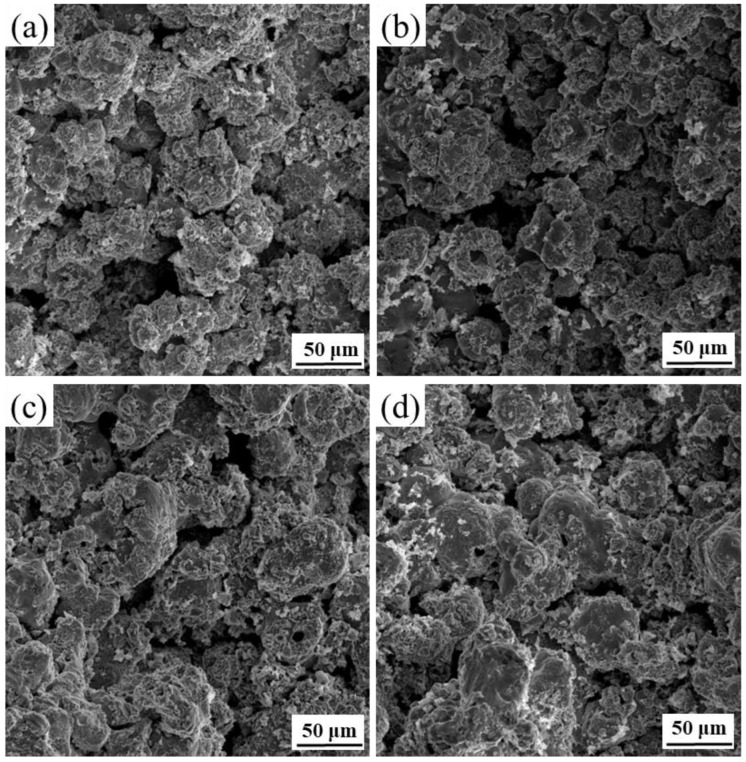
The fracture morphologies of specimens sintered with various solid loading rates: (**a**) 42 vol.%; (**b**) 45 vol.%; (**c**) 48 vol.%; (**d**) 51 vol.%.

**Table 1 materials-15-08398-t001:** The reagents of the gel system used.

Solvent	Organic Monomer	Crosslinker	Dispersant	Initiator	Catalyst
N-octanol	Hydroxyethyl methacrylate (HEMA)	1,6-hexanediol diacrylate (HDDA)	Silok-7456F	Tert-butyl peroxybenzoate (TBPB)	N,N-dimethylaniline (DMA)

## Data Availability

All the data available in main text.

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
