# Peer review of "Influence of Solid Loading on the Gel-Casting of Porous NiTi Alloys"

_materials, 2022, doi:10.3390/ma15238398_

Round 1

Reviewer 2 Report

1) Kindly please enhance the language standard

2) Expand the literature some more details. 

for porosity analysis you may refer " Thermal and mechanical behaviour of sub micron sized fly ash reinforced polyester resin composite"

3) What is exact novelty of the study?

4) Provide equipment manufacture name and location wherever is possible

5) Check unit notation format of Unit

6) Figure quality may be improved

7) Refine the conclusion section

Reviewer 3 Report

The manuscript materials-2024129 was reviewed. The following comments must be addressed:

1. The authors are recommended to mention the other type of metallic foams and porous metallic materials like closed-cell metal foam and metal syntactic foams. To this end, the authors are suggested to use the research studies by N. Movahed et al, E. Liul et al, I.N. Orbulov et al.

2. The methodology for manufacturing the samples must be modified. To this ends, the authors must prepare a schematic presentation for each step of the manufacturing.

3. The manufactured samples must be presented in an individual figure.

4. The shape memory effect of the manufacrtured porous NiTI alloys have not mentioned. It is important to focus to this property and compare it with the developed similar alloys by other research groups.

5. The mechanical properties of the alloy must be calculated using ISO-13314 standard.

Round 2

Reviewer 1 Report

Comments were accepted and the manuscript can be published.

Reviewer 2 Report

The comments have been addressed.

Reviewer 3 Report

The manuscript can be published after editor's final decision.